# Prioritization of neglected tropical zoonotic diseases: A one health perspective from Tigray region, Northern Ethiopia

Tadesse Teferi Mersha[1], Biruk Mekonnen Wolde🄳[2]*, Nigus Abebe Shumuye[1,3], Abrha Bsrat Hailu🄳[2], Abrahim Hassen Mohammed[4], Yisehak Tsegaye Redda[2], Birhanu Hadush Abera[1], Habtamu Taddele Menghistu🄳[2,5]

1 Department of Veterinary Clinical Medicine and Epidemiology, College of Veterinary Sciences, Mekelle University, Mekelle, Tigray, Ethiopia, 2 Department of Basic and Diagnostic Sciences, College of Veterinary Sciences, Mekelle University, Mekelle, Tigray, Ethiopia, 3 Lanzihou Veterinary Research Institute, CAAS, Lanzhou, China, 4 Health Bureau, Research, Projects, and International Relations, National Regional State of Tigray, Mekelle, Ethiopia, 5 Institute of Climate and Society, Mekelle University, Mekelle, Tigray, Ethiopia

* mbiruk1972@gmail.com, biruk.mekonnen@mu.edu.et

**Data Availability Statement:** All relevant data are within the manuscript and its S1, S2 Figs, S1–S6 Files.

## Abstract

Neglected tropical zoonotic diseases (NTZDs) continue to have a major effect on the health of humans and animals. In this study, a one health approach was used to prioritize and rank neglected tropical zoonotic diseases at the regional and zonal levels in Tigray National Regional State, Ethiopia. For prioritization of NTZDs a cross-sectional study through a structured questionnaire was administered to 313 health experts from human and animal health sectors. In addition, focus group discussions (FGD) were held with purposively selected key informants. Descriptive, and Multivariable analysis was applied to report the results and a ranked list of diseases was developed at the zonal and regional level. In the region, 8 of the 12 World Health Organization listed NTZDs were considered major diseases including anthrax, brucellosis, bovine tuberculosis, taeniasis, leishmaniasis, rabies, schistosomiasis, and soil-transmitted helminths. Considering the zoonotic and socioeconomic importance of the diseases at the regional level, rabies ranked 1st whereas anthrax, bovine tuberculosis, leishmaniasis, and brucellosis were ranked from 2nd to 5th, respectively. The FGD result also supported the prioritization result. The Multivariable analysis showed a statistically significant difference in the zonal distribution of anthrax ( = 0.009, OR = 1.16), taeniasis (p<0.001, OR = 0.82), leishmaniasis (p<0.001, OR = 1.91), rabies (p = 0.020, OR = 0.79) and soil-transmitted helminths (p = 0.007, OR = 0.87) but not for brucellosis (p = 0.585), bovine tuberculosis (p = 0.505), and schistosomiasis (p = 0.421). Anthrax (p<0.001, OR = 26.68), brucellosis (p<0.001, OR = 13.18), and taeniasis (p<0.001, OR = 6.17) were considered as the major zoonotic diseases by veterinary practitioners than human health practitioners whereas, leishmaniasis was recognized as a major health challenge by human health professionals. Understanding the priority diseases in the region is supportive for informed decision-making and prioritizes the limited resources to use. Furthermore, strengthening the collaboration between human and animal health professions is important to control the diseases.

**Funding:** This study received financial support from the MSc thesis research support of Mekelle University (MU) through the recurrent budget and MU-NMBU NORAD project (award recipient; TTM).

**Competing interests:** The authors have declared that no competing interests exist.

## Introduction

The increasing interactions of humans and animals within the environment are aggravating the ongoing transmission of zoonoses from cattle to humans and vice versa [1, 2]. Zoonoses are exerting a significant burden on both animal and human health, particularly in developing countries. Zoonotic diseases impose a double burden on the well-being of people by compromising the health and productivity of their livestock; however, they are often neglected by health facility managers and policymakers of the developed and developing world [3].

Neglected tropical zoonotic diseases (NTZDs) are a subset of zoonoses that primarily affect the world's poorest population [2, 4]. The World Health Organization (WHO) has identified NTZDs namely anthrax, bovine tuberculosis (BTB), brucellosis, leptospirosis, rabies, echino-coccosis, food-borne trematodes, human African trypanosomiasis, taeniasis/cysticercosis, and leishmaniasis [2, 3]. Additionally, soil-transmitted helminths and schistosomiasis are included as major neglected zoonotic diseases in Ethiopia [5, 6].

Ethiopia reported the second-highest burden of zoonotic diseases in Africa [7]; yet, NTZDs has not received attention at various levels in the country. Moreover, data on the burden and distribution of these diseases are incomplete and not updated periodically. To date, prevention and control of NTZDs are challenged by the lack of coordinated efforts between human and animal health professionals and other concerned authorities [6]. As a result, NTZDs continued to affect the livelihoods of poor communities. Considering this fact, Ethiopia has developed a multi-year national master plan that enhances the prevention, control, and eradication of neglected tropical diseases [5, 8].

Selection and prioritization of NTZDs are essential to allocate resource-based investment for their control and prevention. Prioritization can also help to identify vulnerabilities not only where zoonosis poses a significant health threat but also where efforts can be focused to improve prevention, communication, and coordination across veterinary and human health [9–11]. In addition to this, understanding the perceptions of human health and veterinary experts on various NTZDs and their risk is a crucial step to properly plan, manage, and monitor any public health system. This facilitates the prediction of zoonotic disease and in turn, guides federal and regional authorities in decision-making and policy planning for cost-effective resource allocation. The one health approach, based on a multi-sectoral collaboration and coordination, plays a significant role in the prevention and control of zoonoses [3].

A study conducted for the prioritization of zoonotic diseases in Ethiopia has identified rabies, anthrax, brucellosis, leptospirosis, and echinococcosis as the top five diseases [12]. However, this study does not show the priority zoonotic diseases at regional and zonal levels where the distribution and burden of the diseases vary with the diverse agro-ecology of the country and potentially different health systems. Moreover, this study was based on the opinion of experts at the federal level and it doesn't take into account the facts on the ground. A regional/ zonal level prioritization of zoonotic diseases is a crucial step for informed decision-making and to design and enforce locally feasible disease control and prevention options. Therefore, this study used a one health approach as a tool to prioritize NTZDs and develop a ranked list of these diseases at the regional and zonal administrative levels in Tigray National Regional State, Ethiopia.

## Materials and methods

### Ethical statement

The study was reviewed and approved by the government of the National Regional State of Tigray, Bureau of Health (Ref.No. 31/1418/17). Permission was also obtained from each study

district and verbal consent was obtained from all volunteer participants. Confidentiality for all collected information was preserved using secret codes for each participant.

## Study area description

The study was conducted from November 2017 to May 2018 in Tigray National Regional State, Ethiopia. Tigray Region is in the northernmost of the country bordered by Eritrea to the north, Sudan to the west, the Afar region to the east, and the Amhara region to the south and southwest. The region is situated between 12˚30'N and 15˚N latitude, and 36˚30'E and 40˚30'E longitude. Topographically the region has a diversified agro-ecology, having an altitude ranging from 500–3,935 meter above sea level. Moreover, the region is characterized by an arid and semi-arid climate with low and erratic rainfall. The mean annual temperature of the region is between 15˚C and 21˚C. The estimated population projections for 2017 based on the 2007 census is 5,247,005 (49.27% male, and 50.73% female) [13]. During this period, the region had seven zones, and 35 districts [14] (Fig 1).

According to the Central Statistical Agency (CSA) projection, the estimated population of the region in 2017 was 5,247,005 (49.27% male, and 50.73% female) [15]. The livestock population of the region was estimated at 4,791,341 heads of cattle, 2,041,731 heads of sheep, 4,584,138 heads of goat, 3,815 heads of horse, 7,634 heads of mule, 838,053 heads of donkey, 54,348 heads of camel, 5,735,973 heads of poultry, and 287,135 beehives [16]. The region has one referral hospital, 16 zonal hospitals, 22 primary hospitals, 202 health centers, 712 health posts, and 159 Veterinary clinics [14].

## Study design, study population and sample size determination

Cross-sectional study design was employed. Key informants' in-depth interview (KIDI) and Focus Group Discussion (FGD) were conducted with health experts (human health and veterinary) from all zones of the region. Depending on their size, infrastructures, and transport accessibility for data collection, one to three districts (a total of 15 districts) were selected from each zone. In addition, the 13 major urban centers in the study zones were considered. In each district/urban center, health experts from human health facilities and veterinary clinics were

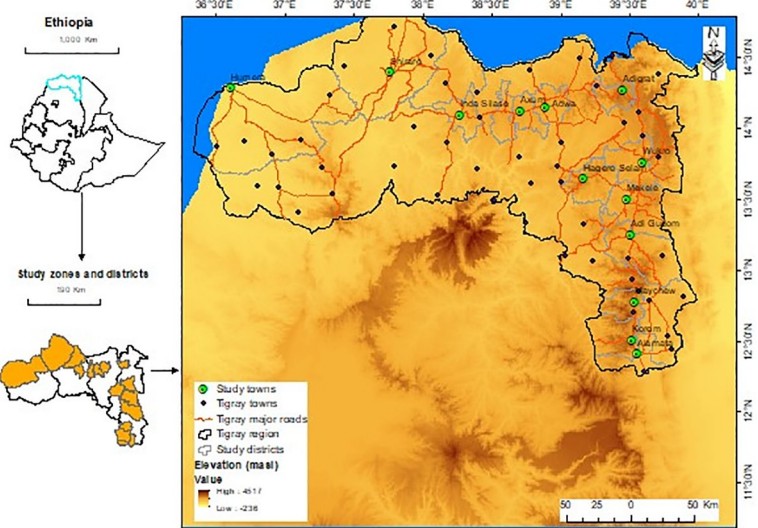

**Fig 1. Map of the study zones, districts, and urban centers.**

selected for the KIDI. Some of them participated in the FGD. For the KIDI, the sample size was calculated based on the formula developed by WHO for questionnaire-based studies in health research and previously used by [17], using n = $Z^{2*}P$ (1-P)/$d^2$ where n = number of health experts (both human health and veterinary experts) enrolled in the study, Z = represents a critical value for the 95% confidence interval (1.96), d = the level of precision (5%) and P = proportion used on expected prevalence (p = 0.5). Since the total health experts in the region were less than 10,000, a correction formula [nf = ni/(1 + ni/N)] was used, where N = total health experts in the region (study population, N = 7278). The number of health experts, in each zone, to be involved in the study was determined using probability proportion (ni = Ni*n/N), where ni = total number of study subjects in each zone, Ni = total number of health experts in each zone, n = total number of study subjects obtained and N = total number of health experts in the region. A 10% non-response rate was also considered to maximize precision. Therefore, 421 health experts were included for the KIDI (Table 1). For the FGD, Health experts from each district were selected purposively based on their understanding and relevance to the study objectives as verified in the KIDI.

## Methods of data collection

**Key informants' in-depth interview (KIDI).**   Key informants were selected from diverse backgrounds (multi-sectoral) including public health, epidemiologist, laboratory technologists, nurses, physicians, disease surveillance experts, and veterinary professionals. Participants were informed of the procedures and significance of the study. A pre-test questionnaire was administered to collect relevant data and information regarding the status of the twelve NTZDs and their impact on the community. Key informants were asked about the severity of illness in humans and/or animals, mode of transmission, impact (health and economic burden) of the diseases, inter-sectoral collaborations, control measures (availability of interventions), and major risk factors of NTZDs (S1 File). The questionnaire survey was pre-tested in two selected districts within two selected study zones. From each district ten professionals (veterinary and human health) were involved in the pre-testing. Based on the results of the pre-test, the questionnaire contents were modified. To minimize bias during the discussion with informants, the information was correlated with data generated from the bureau of health and literature [12, 18, 19].

**Table 1. Number of health experts by zone and sample size determination for questionnaire survey.**

| Zones | Health experts | | | | Animal Health experts | | | | Total sample size |
|---|---|---|---|---|---|---|---|---|---|
| | Total population | ni = (Ni*n/N) | nf = ni/ (1+ni/N) | Sample size with 10% NRR | Total population | ni = (Ni*n/N) | nf = ni/(1+ni/N) | Sample size with 10% NRR | |
| Western | 981 | 51 | 51 | 56 | 82 | 4 | 4 | 5 | 61 |
| Eastern | 1331 | 69 | 69 | 76 | 69 | 4 | 4 | 4 | 80 |
| Southern | 1344 | 70 | 69 | 76 | 74 | 4 | 4 | 4 | 80 |
| N/ Western | 860 | 45 | 45 | 49 | 115 | 6 | 6 | 7 | 56 |
| S/Eastern | 641 | 33 | 33 | 37 | 46 | 2 | 2 | 3 | 40 |
| Mekelle | 448 | 23 | 23 | 26 | 23 | 1 | 1 | 1 | 27 |
| Central | 1237 | 64 | 64 | 70 | 122 | 6 | 6 | 7 | 77 |
| **Total** | **6842** | **390** | | | **531** | **31** | | | **421** |

NRR = Non-response rate; N/Western = Northwestern; S/Eastern = Southeastern.

**Focus group discussion (FGD).** The FGD aimed to support the disease distribution, burden, and prioritization of NTZDs, and to assess the associated risk factors and the laboratory infrastructures in the respective zones. In each zone, a group having 7–12 participants from the selected study districts and urban centers was formed composing the diverse background of experts who already participated in KIDI. The FGD points were mainly focusing on the impact of NTZDs on socioeconomic and public health importance, distribution, and burden of NTZDs in the study zone. Moreover, questions related to the identification of major NTZDs, the status of laboratory infrastructure, diagnostic techniques and treatment efficacy, disease reporting system, inter-sectoral collaboration, and major risk factors of NTZDs were included (S2 File). The discussion was conducted using the local language (Tigrigna) and the researcher took notes and tape-recorded. Finally, the discussion points were transcribed and translated into the English language.

**Prioritization and ranking of neglected tropical zoonotic diseases.** The prioritization started with the identification of a specific list of NTZDs [12] followed by the development of structured ranking criteria [6]. The criteria used for the prioritization include severity of illness to humans (max score = 1), transmission potential between humans and animals (max score = 0.85), economic burden of the disease (max score = 0.65), intersectoral collaboration (max score = 0.45) and availability of intervention (max score = 0.30). If a disease is of high priority, it will get a maximum score of 3.25 for a single respondent. The structured questionnaire was distributed to the selected key informants based on the inclusion criteria (S3 File). The participants were eligible if they had two and above years of work experience in the study area and if they were able to answer at least four questions on knowledge assessment criteria [20, 21]. Finally, the score given by the informants was summarized, and diseases were ranked.

## Data management and analysis

For the in-depth interviews, all the responses of the informants were assessed in relation to the facts obtained from publications and the general truth of NTZDs. The answer for each question was ranked based on their significance of measurement for NTZDs in the area and scores assigned based on the response to each question. A decision tree was designed using Microsoft Excel and used to determine the final disease ranking. The scores for a single question were multiplied by the number of respondents who correctly answered the questions. Finally, the total scores for each disease at the zonal and regional level were summed up to rank the disease according to its priority. Data compiled during the literature review was used to determine appropriate responses for each question for all NTZDs under consideration.

Data obtained from the questionnaire survey were entered into Microsoft Excel 2010 spreadsheet, exported to STATA version 15.0 for Windows (Stata Corp. College Station, USA), coded and analyzed. Descriptive statistics and Multivariable analysis were applied to present the results. A univariable logistic regression was applied to measure the strength of association between the dependent and independent variables before running the Multivariable logistic regression. The logistic regression model was fitted with individual NTZD result in the zones (whether a major zoonoses or not) as the outcome. The model was built using the forward stepwise (conditional) selection procedure by applying the iterative maximum likelihood estimation procedure, while the statistically significant contribution of individual predictors to the models was tested using the Wald's test and likelihood-ratio tests. Any interaction between variables was assessed by constructing a multivariable model as described previously [22, 23]. The logistic model was checked for goodness-of-fit using the Hosmer and Lemeshow test. The odd ratio at 95% CI was computed and results were considered significant at p-value<0.05. Zone, profession, and experience were considered as independent variables against the existence of major NTZDs in the zones.

## Results

### Key informants' in-depth interview (KIDI)

A total of 421study participants were enrolled for KIDI, but 21 of them were excluded due to the incompleteness of their data. Thus, the KIDI analysis was performed based on the response from 400 experts.

### The burden of NTZDs by expert perspectives

Of the 12 listed NTZDs, eight of them namely anthrax, brucellosis, bovine tuberculosis (BTB), taeniasis, leishmaniasis, rabies, schistosomiasis, and soil-transmitted helminths (STH) were considered as the major zoonotic diseases in the region. Rabies (92%, 368/400) followed by BTB (57.8, 231/400), and leishmaniasis (44.5, 178/400) were considered as the most important major zoonotic diseases in the region (Table 2). The ranking of NTZDs by profession showed that anthrax, brucellosis, taeniasis/cysticercosis and hydatidosis were ranked as major NTZDs by veterinarians whereas leishmaniasis, schistosomiasis, and STH were considered as major priority NTZDs by human health experts (S1 Fig). Professionals above 1 year of experience consider NTZDs as major zoonoses in their zones (S2 Fig).

**Table 2. Zonal distribution of major NTZDs and comparisons with profession and working experience.**

| Variable | Category | Anthrax, Yes (%) | Brucella, Yes (%) | BTB, Yes (%) | Taenia, Yes (%) | Hydatid, Yes (%) | Leishmania, Yes (%) | HAT, Yes (%) | Lepto, Yes (%) | Rabies, Yes (%) | Schisto, Yes (%) | STH, Yes (%) | FBT, Yes (%) |
|---|---|---|---|---|---|---|---|---|---|---|---|---|---|
| Zone | WZ (n = 59) | 27 (45.8) | 20 (33.9) | 37 (62.7) | 14 (23.7) | 8 (13.6) | 54 (91.5) | 1 (1.7) | 2 (3.4) | 51 (86.4) | 8 (13.6) | 9 (15.3) | 5 (8.5) |
| | NWZ (n = 53) | 25 (47.2) | 11 (20.8) | 35 (66.0) | 21 (39.6) | 3 (5.7) | 44 (83.0) | 0 (0) | 0 (0) | 48 (90.6) | 27 (50.9) | 15 (28.3) | 2 (3.8) |
| | CZ (n = 75) | 27 (36.0) | 21 (28.0) | 41 (54.7) | 15 (20.0) | 4 (5.3) | 40 (53.3) | 0 (0) | 1 (1.3) | 68 (90.7) | 52 (69.3) | 17 (22.7) | 3 (4.0) |
| | EZ (n = 73) | 40 (54.8) | 19 (26.0) | 32 (43.8) | 27 (37.0) | 5 (6.8) | 11 (15.1) | 2 (2.7) | 3 (4.1) | 65 (89.0) | 12 (16.4) | 40 (54.8) | 20 (27.4) |
| | SEZ (n = 38) | 7 (18.4) | 13 (34.2) | 27 (71.1) | 15 (39.5) | 12 (31.6) | 2 (5.3) | 0 (0) | 6 (15.8) | 38 (100) | 13 (34.2) | 17 (44.7)) | 14 (36.8) |
| | MZ (n = 26) | 10 (38.5) | 6 (23.1) | 19 (73.1) | 16 (61.5) | 8 (30.8) | 16 (61.5) | 3 (11.5) | 9 (34.6) | 26 (100) | 15 (57.7) | 13 (50.0) | 2 (7.7) |
| | SZ(n = 76) | 21 (27.6) | 24 (31.6) | 40 (52.6) | 35 (46.1) | 10 (13.2) | 11 (14.5) | 1 (1.3) | 2 (2.6) | 72 (94.7) | 29 (38.2) | 23 (30.3) | 14 (18.4) |
| Region | | 157(39.3) | 114(28.5) | 231 (57.8) | 143(35.8) | 50(12.5) | 178(44.5) | 7(1.8) | 23(5.8) | 368 (92.0) | 156(39) | 134 (33.5) | 60(15) |
| Profession | Vets (n = 43) | 40 (93.0) | 34 (79.1) | 28 (65.1) | 31 (72.1) | 27 (62.8) | 15 (34.9) | 1 (2.3) | 4 (9.3) | 41 (95.3) | 12 (27.9) | 10 (23.3) | 12 (27.9) |
| | Human (357) | 117 (32.8) | 80 (22.4) | 203 (56.9) | 112 (31.4) | 23 (6.4) | 163 (45.7) | 6 (1.7) | 19 (5.3) | 327 (91.6) | 144 (40.3) | 124 (34.7) | 48 (13.4) |
| Experience | < 1 yr (n = 73) | 21 (28.8) | 16 (21.9) | 40 (54.8) | 24 (32.9) | 6 (8.2) | 33 (45.2) | 0 (0) | 2 (2.7) | 62 (84.9) | 23 (31.5) | 23 (31.5) | 12 (16.4) |
| | 1–3 yrs (n = 82) | 29 (35.4) | 26 (31.7) | 47 (57.3) | 28 (34.1) | 7 (8.5) | 41 (50.0) | 3 (3.7) | 4 (4.9) | 76 (92.7) | 33 (40.2) | 22 (26.8) | 15 (18.3) |
| | 3–5 yrs (n = 87) | 31 (35.6) | 23 (26.4) | 55 (63.2) | 30 (34.5) | 12 (13.8) | 33 (37.9) | 0 (0) | 8 (9.2) | 86 (98.9) | 32 (36.8) | 32 (36.8) | 14 (16.1) |
| | > 5 yrs (n = 158) | 76 (48.1) | 49 (31.0) | 89 (56.3) | 61 (38.6) | 25 (15.8) | 71 (44.9) | 4 (2.5) | 9 (5.7) | 144 (91.1) | 86 (54.4) | 57 (36.1) | 19 (12.0) |

**NB**: Brucella = Brucellosis; BTB = Bovine Tuberculosis; Taenia = Taeniasis; Hydatid = Hydatidosis; Leishamania = Leishmaniasis; HAT = Human African Trypanosomiasis; Lepto = Leptospirosis; Schisto = Schistosomiasis; STH = Soil Transmitted Helminths; FBT = Foodborne Trematodes.

WZ = Western zone; NWZ = Northwestern zone; CZ = Central zone; EZ = Eastern zone; SEZ = Southeastern zone; MZ = Mekelle zone; SZ = Southern zone.

**Table 3. Multivariable analysis of anthrax, brucellosis, TB, and taeniasis as major zoonoses against the zone, profession, and work experience.**

| Variable | Category | Anthrax | | Brucellosis | | Tuberculosis | | Taeniasis/cysticercosis | |
|---|---|---|---|---|---|---|---|---|---|
| | | Yes (%) | AOR (95%CI) | Yes (%) | AOR (95%CI) | Yes (%) | OR (95%CI) | Yes (%) | OR (95%CI) |
| Zone | WZ (n = 59) | 27 (45.8) | | 20 (33.9) | | 37 (62.7) | | 14 (23.7) | |
| | NWZ (n = 53) | 25 (47.2) | 1.184 (0.519–2.702) | 11 (20.8) | 2.768 (1.032–7.422) | 35 (66.0) | 0.884 (0.402–1.942) | 21 (39.6) | 0.461 (0.193–1.099) |
| | CZ (n = 75) | 27 (36.0) | 1.585 (0.738–3.404) | 21 (28.0) | 1.234 (0.548–2.778) | 41 (54.7) | 1.371 (0.679–2.768) | 15 (20.0) | 1.190 (0.449–2.837) |
| | EZ (n = 73) | 40 (54.8) | 0.676 (0.320–1.426) | 19 (26.0) | 1.409 (0.613–3.237) | 32 (43.8) | 2.121 (1.043–4.315) | 27 (37.0) | 0.474 (0.210–1.067) |
| | SEZ (n = 38) | 7 (18.4) | 4.832 (1.585–14.728) | 13 (34.2) | 0.854 (0.332–2.193) | 27 (71.1) | 0.719 (0.294–1.757) | 15 (39.5) | 0.379 (0.148–0.972) |
| | MZ (n = 26) | 10 (38.5) | 1.534 (0.536–4.384) | 6 (23.1) | 1.758 (0.533–5.799) | 19 (73.1) | 0.625 (0.224–1.745) | 16 (61.5) | 0.164 (0.058–0.464) |
| | SZ (n = 76) | 21 (27.6) | 2.643 (1.180–5.922) | 24 (31.6) | 1.086 (0.488–2.414) | 40 (52.6) | 1.572 (0.779–3.173) | 35 (46.1) | 0.309 (0.139–0.685) |
| Profession | Vets (n = 43) | 40 (93.0) | | 34 (79.1) | | 28 (65.1) | | 31 (72.1) | |
| | Human (357) | 117 (32.8) | 30.801 (9.065–104.648) | 80 (22.4) | 15.163 (6.732–34.153) | 203 (56.9) | 1.419 (0.718–2.806) | 112 (31.4) | 6.258 (2.986–13.114) |
| Experience | < 1 yr(n = 73) | 21 (28.8) | | 16 (21.9) | | 40 (54.8) | | 24 (32.9) | |
| | 1–3 yrs(n = 82) | 29 (35.4) | 0.593 (0.280–1.256) | 26 (31.7) | 0.513 (0.235–1.121) | 47 (57.3) | 0.951 (0.494–1.830) | 28 (34.1) | 0.959 (0.468–1.964) |
| | 3–5 yrs(n = 87) | 31 (35.6) | 0.560 (0.262–1.119) | 23 (26.4) | 0.807 (0.358–1.816) | 55 (63.2) | 0.802 (0.414–1.554) | 30 (34.5) | 1.276 (0.619–2.629) |
| | > 5 yrs (n = 158) | 76 (48.1) | 0.510 (0.261–0.997) | 49 (31.0) | 0.694 (0.338–1.427) | 89 (56.3) | 1.016 (0.569–1.815) | 61 (38.6) | 0.974 (0.515–1.841) |

**NB:** WZ = Western zone; NWZ = Northwestern zone; CZ = Central zone; EZ = Eastern zone; SEZ = Southeastern zone; MZ = Mekelle zone; SZ = Southern zone.

According to the perception of experts using the Multivariable analysis, anthrax was considered 4.83 and 2.64 times as a major zoonosis in Western zone compared to Southeastern (AOR = 4.832, CI = 1.585–14.728) and Southern (AOR = 4.643, CI = 1.180–5.922) zones, respectively; whereas brucellosis was 2.77 times more important zoonotic diseases in Western zone (AOR = 2.768, CI = 1.032–7.422) than Northwestern zone. Bovine TB was also considered 2.12 times as a major zoonosis in Western zone compared to Eastern zone (AOR = 2.121, CI = 1.043–4.315). Taeniasis was considered as less major zoonoses in Western zone of the region compared to Southeastern, (AOR = 0.379, CI = 0.148–0.972), Mekelle (AOR = 0.164, CI = 0.058–0.464), and Southern (AOR = 0.308, 0.139–0.685) zones (Table 3).

Leishmaniasis was considered as a major zoonosis by 91.5% and 83% of the experts in Western and North-western zones, respectively, and the disease was considered 10.103, 66.464, 229.974, 6.850, and 74.936 times as major zoonosis in the Western zone compared to Central(AOR = 10.103, CI = 3.544–28.804), Eastern(AOR = 66.464, CI = 21.135–209.012), Southeastern(AOR = 229.974, CI = 40.930–1292.138), Mekelle (AOR = 6.850, CI = 1.978–23.728) and Southern(AOR = 74.936, CI = 23.676–237.180) zones, respectively. Rabies was considered as a major zoonosis in all zones of the region and there was no statistically significant variation between zones. According to the perception of experts, the burden of schistosomiasis was less in the Western zone by 0.16, 0.07, 0.30, 0.11, and 0.25 times, respectively, than Northwestern(AOR = 0.159, CI = 0.062–0.404), Central(AOR = 0.069, CI = 0.028–0.170), Southeastern(AOR = 0.302, CI = 0.109–0.840), Mekelle(AOR = 0.112, CI = 0.037–0.337), and

**Table 4. Multivariable analysis of leishmaniasis, rabies, schistosomiasis and STH as major zoonoses against the zone, profession, and work experience.**

| Variable | Category | Leishmaniasis | | Rabies | | Schistosomiasis | | STH | |
|---|---|---|---|---|---|---|---|---|---|
| | | Yes (%) | AOR (95%CI) | Yes (%) | AOR (95%CI) | Yes (%) | OR (95%CI) | Yes (%) | OR (95%CI) |
| Zone | WZ (n = 59) | 54 (91.5) | | 51 (86.4) | | 8 (13.6) | | 9 (15.3) | |
| | NWZ (n = 53) | 44 (83.0) | 2.087 (0.635–6.864) | 48 (90.6) | 0.869 (0.252–2.999) | 27 (50.9) | 0.159 (0.062–0.404) | 15 (28.3) | 0.467 (0.182–1.97) |
| | CZ (n = 75) | 40 (53.3) | 10.103 (3.544–28.805) | 68 (90.7) | 0.625 (0.207–1.890) | 52 (69.3) | 0.069 (0.028–0.170) | 17 (22.7) | 0.650 (0.264–1.597) |
| | EZ (n = 73) | 11 (15.1) | 66.464 (21.135–209.012) | 65 (89.0) | 0.759 (0.257–2.238) | 12 (16.4) | 0.840 (0.315–2.242) | 40 (54.8) | 0.157 (0.067–0.368) |
| | SEZ (n = 38) | 2 (5.3) | 229.974 (40.930–1292.138) | 38 (100) | 1.000 | 13 (34.2) | 0.302 (0.109–0.840) | 17 (44.7)) | 0.238 (0.090–0.628) |
| | MZ (n = 26) | 16 (61.5) | 6.850 (1.978–23.728) | 26 (100) | 1.000 | 15 (57.7) | 0.112 (0.037–0.337) | 13 (50.0) | 0.194 (0.067–0.560) |
| | SZ (n = 76) | 11 (14.5) | 74.936 (23.676–237.180) | 72 (94.7) | 0.434 (0.121–1.556) | 29 (38.2) | 0.248 (0.102–0.603) | 23 (30.3) | 0.420 (0.176–1.000) |
| Profession | Vets (n = 43) | 15 (34.9) | | 41 (95.3) | | 12 (27.9) | | 10 (23.3) | |
| | Human (357) | 163 (45.7) | 0.340 (0.140–0.824) | 327 (91.6) | 2.228 (0.495–10.034) | 144 (40.3) | 0.492 (0.230–1.057) | 124 (34.7) | 0.540 (0.248–1.176) |
| Experience | < 1 yr(n = 73) | 33 (45.2) | | 62 (84.9) | | 23 (31.5) | | 23 (31.5) | |
| | 1–3 yrs(n = 82) | 41 (50.0) | 0.846 (0.354–2.021) | 76 (92.7) | 0.413 (0.140–1.221) | 33 (40.2) | 0.605 (0.287–1.279) | 22 (26.8) | 1.165 (0.557–2.437) |
| | 3–5 yrs(n = 87) | 33 (37.9) | 0.948 (0.400–2.247) | 86 (98.9) | 0.081 (0.010–0.659) | 32 (36.8) | 0.791 (0.377–1.659) | 32 (36.8) | 0.859 (0.424–1.742) |
| | > 5 yrs (n = 158) | 71 (44.9) | 1.166 (0.558–2.435) | 144 (91.1) | 0.570 (0.234–1.392) | 86 (54.4) | 0.576 (0.295–1.125) | 57 (36.1) | 0.852 (0.452–1.608) |

NB:WZ = Western zone; NWZ = Northwestern zone; CZ = Central zone; EZ = Eastern zone; SEZ = Southeastern zone; MZ = Mekelle zone; SZ = Southern zone.

Southern(AOR = 0.248, CI = 0.102–0.603) zones. Similarly, the burden of STH was less in the Western zone by 0.16, 0.24, and 0.19 times, respectively, compared to Eastern (AOR = 0.157, CI = 0.067–0.368), Southeastern(AOR = 0.237, CI = 0.090–0.628), and Mekelle(AOR = 0.194, CI = 0.067–0.560) zones (Table 4).

Compared to health professionals, veterinarians indicated anthrax, brucellosis, and taeniasis 30.80(AOR = 30.801, CI = 9.065–104.648), 15.16(AOR = 15.163, CI = 6.732–34.153), and 6.26 (AOR = 6.258, CI = 2.986–13.114) times, respectively, as major health challenges in their professional carrier (Table 3). However, leishmaniasis was considered as less health challenge by veterinary professionals (AOR = 0.340, CI = 0.140–0.824) than human health experts (Table 4), but it was identified as an important zoonosis by human health experts. Except for the case of rabies, the experience of the health experts didn't influence to consider these diseases as major zoonoses in their respective zones. However, health experts with experience less than 1 year considered rabies 0.08 (AOR = 0.081, CI = 0.010–0.659) times less than experts with 3–5 years' experience as a major zoonosis (Table 4).

## Prioritization of NTZDs by health experts

Out of the 400 interviewed respondents, considering the identified exclusion criteria, responses from 313 experts were considered for the disease ranking. Rabies ranked 1st with a regional score of 885.83 out of the maximum score of 1017.25. Anthrax, BTB, leishmaniasis, brucellosis, and leptospirosis ranked from 2nd to 6th, respectively (Table 5). Likewise, the three major NTZDs in each zone were rabies, anthrax, and BTB (Table 6).

**Table 5. Regional prioritization of neglected tropical zoonotic disease by experts.**

| Diseases | Regional Score | Rank |
|---|---|---|
| Rabies | 885.83 | 1st |
| Anthrax | 792.08 | 2nd |
| Bovine Tuberculosis (BTB) | 785.85 | 3rd |
| Leishmaniasis | 724.27 | 4th |
| Brucellosis | 721.97 | 5th |
| Leptospirosis | 642.57 | 6th |
| Hydatidosis | 638.52 | 7th |
| Schistosomiasis | 592.06 | 8th |
| Taeniasis | 571.55 | 9th |
| Food-borne Trematodes (FBT) | 570.05 | 10th |
| Soil-Transmitted Helminths (STH) | 550.68 | 11th |
| Human African Trypanosomiasis (HAT) | 532.86 | 12th |

## Focus group discussion (FGDs)

Concerning the impact of NTZDs on public health and socio-economic importance, the discussants from all zones of the region indicated the adverse effects of NTZDs on both human and animal health. According to the discussants, the major negative impacts of these diseases in animals were high morbidity and mortality, chronic illness, low production, and productivity, international trade restriction, and retard genetic improvement programs. Similarly, the public health effects of NTZDs identified by the discussants were overburden on the public health system, massive economic and social troubles, insufficient human resource development, stunting growth, easy way of disease transmission, and compromising the working performance of individuals. One of the key messages forwarded by the discussants was "*as significant numbers of impoverished people may not afford the additional costs for medications, often they are trapped in a never-ending cycle of poverty. Because of the endemicity of these diseases, there is also an impact on the cost of treatment, control and prevention programs*".

Rabies, bovine tuberculosis, anthrax, leishmaniasis, schistosomiasis, brucellosis were identified as major NTZDs with wider distribution across the region. According to the discussants,

**Table 6. Zonal prioritization of neglected tropical zoonotic disease by experts.**

| Disease | WZ | | NWZ | | CZ | | EZ | | SEZ | | MZ | | SZ | |
|---|---|---|---|---|---|---|---|---|---|---|---|---|---|---|
| | Score | Rank | Score | Rank | Score | Rank | Score | Rank | Score | Rank | Score | Rank | Score | Rank |
| Rabies | 135.18 | 1st | 146.4 | 1st | 160.75 | 1st | 164.05 | 1st | 61.18 | 1st | 51.59 | 1st | 166.64 | 1st |
| Anthrax | 120.69 | 3rd | 130.2 | 2nd | 143.13 | 2nd | 147.99 | 3rd | 55.85 | 2nd | 44.67 | 2nd | 149.60 | 2nd |
| BTB | 125.97 | 2nd | 125.1 | 3rd | 139.87 | 3rd | 152.27 | 2nd | 50.90 | 4th | 43.18 | 3rd | 148.54 | 3rd |
| Leishmaniasis | 113.62 | 4th | 119.1 | 4th | 132.54 | 5th | 131.00 | 5th | 50.00 | 5th | 41.52 | 4th | 136.50 | 4th |
| Brucellosis | 106.25 | 5th | 117.9 | 5th | 135.46 | 4th | 138.63 | 4th | 51.98 | 3rd | 39.37 | 5th | 132.41 | 5th |
| Shistosomiasis | 87.76 | 6th | 99.4 | 6th | 108.88 | 8th | 111.44 | 8th | 43.54 | 8th | 30.67 | 8th | 110.39 | 8th |
| Hydatidosis | 85.15 | 7th | 94.8 | 8th | 118.63 | 7th | 124.93 | 7th | 47.51 | 6th | 36.68 | 7th | 130.82 | 6th |
| Leptospirosis | 84.13 | 8th | 97.4 | 7th | 124.68 | 6th | 125.50 | 6th | 45.48 | 7th | 37.18 | 6th | 128.21 | 7th |
| Taeniasis | 82.23 | 9th | 92.8 | 9th | 106.36 | 10th | 109.90 | 9th | 42.05 | 10th | 29.85 | 9th | 108.38 | 9th |
| STH | 81.60 | 11th | 90.7 | 11th | 104.21 | 11th | 106.67 | 11th | 38.87 | 11th | 29.66 | 11th | 98.93 | 12th |
| HAT | 81.59 | 12th | 88.2 | 12th | 96.72 | 12th | 96.93 | 12th | 38.37 | 12th | 29.20 | 12th | 101.82 | 11th |
| FBT | 81.83 | 10th | 92.6 | 10th | 107.72 | 9th | 107.60 | 10th | 42.74 | 9th | 29.40 | 10th | 108.13 | 10th |

**NB:**WZ = Western zone; NWZ = Northwestern zone; CZ = Central zone; EZ = Eastern zone; SEZ = Southeastern zone; MZ = Mekelle zone; SZ = Southern zone.

**Table 7. List of top five NTZDs by zone through the FGD.**

| Disease rank | Zone | | | | | |
|:---:|:---:|:---:|:---:|:---:|:---:|:---:|
| | **WZ** | **NWZ** | **CZ** | **EZ** | **MZ and SEZ** | **SZ** |
| 1 | Rabies | Rabies | Rabies | Rabies | Rabies | Rabies |
| 2 | Leishmaniasis | Leishmaniasis | Schistosomiasis | Tuberculosis | Anthrax | Tuberculosis |
| 3 | Brucellosis | Anthrax | Anthrax | Anthrax | Helminthiasis | Anthrax |
| 4 | Tuberculosis | Brucellosis | Tuberculosis | Helminthiasis | Tuberculosis | Brucellosis |
| 5 | Anthrax | Tuberculosis | Helminthiasis | Brucellosis | Leishmaniasis | Schistosomiasis |

**NB:** WZ = Western zone; NWZ = Northwestern zone; CZ = Central zone; EZ = Eastern zone; SEZ = Southeastern zone; MZ = Mekelle zone; SZ = Southern zone.

rabies was considered as the most important zoonotic disease with wider distribution throughout the zones (Table 7).

Discussion on disease reporting system and institutional collaboration indicated that there was poor veterinary case recording system in all zones of the region characterized by limited recording practice and disease data management. Rather diseases were reported generally as bacterial infection, parasitic infection, viral infection, and so on. On the other hand, the human health recording and reporting system are much better and are computerized through the Health Management Information System (HMIS) from the bottom (district) to top (region). However, HMIS data does not include disease reports of anthrax, brucellosis, hydatidosis, taeniasis, soil-transmitted helminths, and food-borne trematodes that are listed by WHO and the national master plan of Ethiopia for neglected tropical diseases. All the discussants agreed that the collaboration between human and animal health professionals in disease communication, control, and prevention was weak. Collaborations between the two sectors are initiated whenever there is an outbreak of rabies. Lack of awareness about NTZDs, free animal movement and illegal trade, close human-animal interaction, feeding habits of raw animal product, illegal slaughtering practices, presence of suitable environment for some vector-borne NTZDs, and lack of planned control and prevention strategies and laboratory infrastructure were identified as major risk factors in epidemiology and dynamics of NTZDs.

## Discussion

In the present study, prioritization and ranking of NTZDs were conducted through KIDI and triangulated with the FGDs. Prioritization of NTZDs permits a region to reconsider priority diseases periodically and guides the direction of the limited resource allocation.

In this study, the top five prioritized NTZDs in Tigray region in descending order of importance were rabies, anthrax, bovine tuberculosis (BTB), leishmaniasis, and brucellosis. These top five priority NTZDs were also included in the list of priority zoonotic diseases at the country level, Ethiopia [12] reflecting the importance of these diseases in both the region and the country. In addition, a mapping study conducted in selected districts of Tigray, Afar, and Amhara regions has shown rabies, TB, leishmaniasis, and schistosomiasis as major NTZDs [8]. Rabies, anthrax, and brucellosis were considered in Kenya and Uganda among the top priority zoonotic diseases [24, 25].

Rabies is a fatal but neglected zoonotic disease, which constitutes a major public health concern globally [26, 27]. In this study, the KIDIand FGD results highlighted the importance of rabies in the region, where it was ranked as the 1st priority zoonoses in all the zones and the region. The high rank of rabies in the region could be due to the poor management of owned dogs, the presence of a high population of unvaccinated stray dogs, and lack of an effective

rabies control program [28–31]. This indicates the region is far behind the Global Rabies eradication program by 2030 [32].

Furthermore, anthrax and BTB were prioritized as the other next important NTZDs. Focus group discussion results also support the fact that anthrax and BTB ranked as the top five NTZDs in all zones of the region. Anthrax is among the most prevalent diseases of animals with repeated annual outbreak and is one of the major public health concern in Ethiopia with the highest human case prevalence reported in the Tigray region [5, 33]. Even though there is an effective anthrax vaccine for use in animals, the annual vaccination coverage is challenged by limited vaccine delivery and public reluctance to vaccinate their animals [34]. This suggests the need for increased community awareness on public health importance of anthrax in addition to rising anthrax vaccine coverage in the animals from time to time as human anthrax occurs by contact with infected animals or contaminated animal products [35–37]. The country has no vaccine for use in humans.

Bovine tuberculosis (BTB) is a recognized endemic disease of cattle in Ethiopia and is a frequent cause of zoonotic human tuberculosis. The absence of effective cattle TB control programs and lack of routine milk pasteurization procedures in Ethiopia favors the widespread of human tuberculosis due to *Mycobaterium bovis*(*M. bovis*) [38, 39]. In addition, the close physical contact between farmers and their cattle in the Tigray region [40]promotes aerosol transmission. In human beings, infection with BTB presents with a special challenge for patient treatment and recovery as *M. bovis* is naturally resistant to pyrazinamide, one of the four medications used in the standard first-line anti-tuberculosis treatment regimen leading to long hospitalization, high treatment cost, and low productivity [41].

In this study, leishmaniasis ranked as 4th priority NTZDs in the region but with a statistically significant variation in the zonal distribution. Of the respondents, 91.5%, 83%, and 53.3% of the experts from Western, Northwestern, and Central zones, respectively indicated that the disease is a major zoonosis in their respective zones, which was supported by the FGD results too. This finding is in line with prior researches in different parts of the country [42–44]. The reasons for an increment of the disease in the areas particularly in the Western zone Kafta-Humera could be strongly associated with the endemic nature of the disease, migration of a large number of the labor force for the job opportunity, high temperature, and the environment is very suitable for the survival and replication of the vector [44–46].

Brucellosis is reported in many regions of the world including Ethiopia as endemic with high economic loss and zoonotic potential [47–49]. In the present study, brucellosis ranked as the fifth most important zoonotic disease in the region, which is in line with the findings of previous studies [12, 24, 25]. The high prevalence of the disease in animals and anthropogenic factors such as eating habits, poor hygiene, and practices that expose humans to infected animals or their products play a major role in the epidemiology of the disease [48, 50].

Unlike the reports from neighboring Kenya, Somalia, and Uganda, Human African Trypanosomiasis was ranked as the least priority NTZD in the present study [24, 25, 51]. This could be due to the reason that Tigray region lies outside the tsetse belt areas of the country where there is no risk to acquire the disease [52]. In addition, the disease was not considered among the priority diseases in the national neglected tropical disease in Ethiopia [5]and it was not also among the prioritized diseases according to a previous study at national level [12].

There was a statistically significant difference in the zonal distribution of rabies, anthrax, leishmaniasis, taeniasis, and soil-transmitted helminths. This zonal variation could be due to differences in health infrastructure facilities, agro-ecology that can affect pathogen survival and persistence, availability and implementation of disease management measures, and awareness of communities to these NTZDs.

Anthrax, brucellosis, and taeniasis showed statistically significant variation with the perspectives of professionals. This may be explained by the lack of a reporting system for anthrax, brucellosis, and taeniasis in the HMIS, which downgrades the zoonotic and economic importance of these diseases [8]. On the contrary, these diseases are frequently encountered in veterinary practice. The FGD result also indicated a weak inter-sectoral collaboration between human health and livestock sectors in dealing with the prevention and control of NTZDs which is in agreement with previous studies [2, 3, 12, 53–55].

The negative impacts of NTZDs identified in this study include high morbidity and mortality, chronic illness, low production, and productivity, massive economic and social troubles, high cost of treatment, control and prevention programs, and export-import trade restrictions. These were also indicated as major impacts of zoonoses in previous reports [2, 3, 5, 6, 25, 47, 55]. Lack of awareness about NTZDs, free animal movement and illegal trade, close human-animal interaction, feeding habits of raw animal product, illegal slaughtering practices, suitable environment for some vector-borne NTZDs, lack of planned control and prevention strategies, and lack of laboratory infrastructure were considered as the major risk factors and constraints in relation to NTZDs [2, 5, 35, 55].

This study presented some limitations: First, the absence of context specific to one health zoonotic disease prioritization tool could have influenced the prioritization of diseases. In addition, the lack of country-specific data for the majority of the zoonotic diseases has made it difficult to triangulate the results obtained in this study. Second, despite the fact those participants were from diversified disciplines, the number of animal health experts was fewer than human health experts, which may result in bias on professional opinions. In addition, the difference in the background of professionals involved in the study might have influenced the weighing and scoring results of zoonotic diseases. Third, only 15 accessible districts which had a better road infrastructure and health facility were considered in this study which may not infer the non-accessible districts of the region.

## Conclusion

The prioritization result of the present study indicated rabies, anthrax, bovine tuberculosis, leishmaniasis, and brucellosis as the top five major zoonotic diseases in the region. In all the zones of the region, rabies was ranked as the first priority disease. Even though anthrax and brucellosis are identified as a priority NTZDs, unfortunately, these were not included in the HMIS reporting system. The importance of BTB as a priority disease was identified across all zones and was equally recognized by both human and animal health professionals, and individuals with different working experience. The presence of the diseases could be amplified by limited commitment and policy dialogues to contain, socio-cultural practices, poor diagnostic capacity, and lack of coordinated control and prevention programs.

## Supporting information

**S1 Fig. Rating of neglected tropical zoonotic diseases (NTZD) by profession.**
(TIF)

**S2 Fig. Rating of neglected tropical zoonotic diseases (NTZD) by experience of professionals.**
(TIF)

**S1 File. Questionnaire for professionals.**
(PDF)

**S2 File. FGD discussion points for health professionals (human health and veterinarians).**
(PDF)

**S3 File. Tool for prioritization of NTZD in Tigray region, Northern Ethiopia.**
(PDF)

**S4 File.**
(XLSX)

**S5 File.**
(XLSX)

**S6 File.**
(XLSX)

## Acknowledgments

The authors express their sincere appreciation to Tigray National Regional State Bureau of Health and study site veterinary and health facilities for assistance with obtaining the consent and facilitation during data collection. We also thankfully acknowledge study participants (experts) for their participation. Finally, we thank Mrs. Rahel HMIS data officer at the Tigray Region Bureau of Health for her unreserved support on providing HMIS data necessary for the study. Finally, we also gratefully acknowledge Dr. Matthew Thomas (Associate Professor, College of Veterinary Medicine, Iowa State University) for his help in editing the English language.

## Author Contributions

**Conceptualization:** Biruk Mekonnen Wolde, Nigus Abebe Shumuye, Habtamu Taddele Menghistu.

**Data curation:** Tadesse Teferi Mersha.

**Formal analysis:** Habtamu Taddele Menghistu.

**Funding acquisition:** Tadesse Teferi Mersha.

**Methodology:** Tadesse Teferi Mersha, Nigus Abebe Shumuye, Yisehak Tsegaye Redda, Birhanu Hadush Abera, Habtamu Taddele Menghistu.

**Supervision:** Nigus Abebe Shumuye, Abrha Bsrat Hailu, Abrahim Hassen Mohammed, Habtamu Taddele Menghistu.

**Writing – original draft:** Tadesse Teferi Mersha.

**Writing – review & editing:** Biruk Mekonnen Wolde, Nigus Abebe Shumuye, Abrha Bsrat Hailu, Abrahim Hassen Mohammed, Yisehak Tsegaye Redda, Birhanu Hadush Abera, Habtamu Taddele Menghistu.

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
