## [Decision Letter · Decision Letter 0]

13 Jan 2021

PONE-D-20-30909

Prioritization of Neglected  Tropical Zoonotic Diseases: A One Health Perspective from Tigray Region, Northern Ethiopia

PLOS ONE

Dear Dr. Mekonnen,

Thank you for submitting your manuscript to PLOS ONE. After careful consideration, we feel that it has merit but does not fully meet PLOS ONE’s publication criteria as it currently stands. Therefore, we invite you to submit a revised version of the manuscript that addresses the points raised during the review process.

Please attend to all the comments and concerns that were raised by the reviewers. As mentioned by reviewer #1, please seek help from a native English speaker or an editing service to improve the English language in the manuscript. In addition, please attend to the following: 

a) In line 97 to 103, the subheading reads "Study area and population", however what is written in the text under it describes sampling. Please give a detailed description of the study area and the reference population as indicated in the sub-title.

b) In line 119, Z is not a test statistic. Its represents a critical value for the 95% confidence internal in the normal distribution.

c) Please concisely describe the multivariate regression analysis technique that was used. Also describe the criteria that was used to ensure that the model fitted the data. In addition, please seek assistance from someone knowledgeable about how to interpret  and present odd rations. Its important that these results are properly presented in a table.

d) Indicate the meaning of the abbreviations used for zones as you have done for diseases in Tables 2 and 3 and anywhere else where this applies.

e) Table on page 16 should be Table 4.

We look forward to receiving your revised manuscript.

Kind regards,

Martin Chtolongo Simuunza, PhD

Academic Editor

PLOS ONE

Journal Requirements:

2. Please include additional information regarding the survey or questionnaire used in the study and ensure that you have provided sufficient details that others could replicate the analyses. For instance, if you developed a questionnaire as part of this study and it is not under a copyright more restrictive than CC-BY, please include a copy, in both the original language and English, as Supporting Information. Moreover, please include more details on how the questionnaire was pre-tested, and whether it was validated.

4. Please include a copy of Table 4 which you refer to in your text on page 12 and 13.

6.We note that Figure(s) 1 in your submission contain map images which may be copyrighted. All PLOS content is published under the Creative Commons Attribution License (CC BY 4.0), which means that the manuscript, images, and Supporting Information files will be freely available online, and any third party is permitted to access, download, copy, distribute, and use these materials in any way, even commercially, with proper attribution. For these reasons, we cannot publish previously copyrighted maps or satellite images created using proprietary data, such as Google software (Google Maps, Street View, and Earth). For more information, see our copyright guidelines: http://journals.plos.org/plosone/s/licenses-and-copyright.

a)    You may seek permission from the original copyright holder of Figure(s) 1 to publish the content specifically under the CC BY 4.0 license. 

Reviewers' comments:

Reviewer's Responses to Questions

**Comments to the Author**

1. Is the manuscript technically sound, and do the data support the conclusions?

Reviewer #1: Yes

Reviewer #2: Yes

2. Has the statistical analysis been performed appropriately and rigorously? 

Reviewer #1: Yes

Reviewer #2: Yes

3. Have the authors made all data underlying the findings in their manuscript fully available?

Reviewer #1: Yes

Reviewer #2: Yes

4. Is the manuscript presented in an intelligible fashion and written in standard English?

Reviewer #1: Yes

Reviewer #2: Yes

5. Review Comments to the Author

Reviewer #1: Review comments for PONE -D-20-30909 manuscript

General comments on the manuscript

The manuscript Prioritization of Neglected Tropical Zoonotic Diseases: A One Health Perspective from Tigray Region, Northern Ethiopia, showed important results that would guide in the management, prevention and control of NTZDs. It brought out important diseases as perceived by the health and veterinary officials. It also points out that there is really not much collaboration between the medics and the Vets in the management of these NTZDs, a finding which cuts across many developing countries. More work in terms of advocacy is need for the one health approach to management of diseases of zoonotic nature. That said, the manuscript is well written but is without grammatical errors.

Specific comments on the manuscript.

Line 57: The word Taenia is misspelt. The disease recognized by WHO as a NTZD is taeniasis/cysticercosis. The two go together as most often than note, tapeworm carriers also suffer from cysticercosis as a result of auto-infection. In fact, the disease (cysticercosis) is often diagnosed in animals at postmortem.

Line 108: Replace “that” with “at” in the sentence ………….estimated at and not that.

Line 130: Table 1. Sample size for South-eastern zone should be 40 (37 + 3) instead of 39. Better still the authors should explain why sample size for this particular zone was calculated differently.

Line 301: FGD should be written in full as it is at the beginning of the sentence.

Line 303: A full stop in missing between Region and Anthrax. Rephrase the sentence which is supposed to start now with “Anthrax is ……………………..”

Line 306: …………..public reluctance to vaccinate their animals. The authors need to consider the cost of the vaccine. Are these vaccinations against anthrax organized by government? Are they free vaccination campaigns or is the service free to farmers? The answers to these questions may be the reason the farmers are reluctant to vaccinate against anthrax or any other NTZD.

Line 314: M. bovis should be written in full as it is appearing here for the first time in the manuscript. It is not correct to assume that the reader of the manuscript knows what M. bovis is.

Line 329: Replace “and” with “as” between Ethiopia and endemic.

Line 330 to 331: The sentence beginning “In the present study, brucellosis ………………in other studies [12, 21, 22].

Line 335 to 338: This paragraph should be rephrased. It is difficult to comprehend what the authors are trying to put across.

Line 345: Delete ”fact that” in the sentence.

Reviewer #2: This is an informative paper that analyzed survey data in a one health approach from veterinarians and public health officials to evaluate the most important zoonoses across Northern Ethiopia. The article showed differences regionally in the importance of different zoonoses and between the most important pathogens according to the type of practitioner (veterinary vs human). Table 2 is a bit overwhelming and you could perhaps do barplots for differences in disease type and profession and experience and profession. This article I think is useful information for the zoonotic disease literature in terms of understanding expert's perception on the most significant diseases in their region.

Minor comments (there might be some minor additional English errors)

plosone-

line 86 put 'a' before 'one health approach'

lines 108-109 - no need to capitalize the animals

6. PLOS authors have the option to publish the peer review history of their article (what does this mean?). If published, this will include your full peer review and any attached files.

Reviewer #1: **Yes: **Chummy Sikalizyo Sikasunge

Reviewer #2: No

---

## [Author Response · Author response to Decision Letter 0]

17 May 2021

Date: 5th May 2021

Ref: PONE-D-20-30909

Manuscript Title: “Prioritization of Neglected Tropical Zoonotic Diseases: A One Health Perspective from Tigray Region, Northern Ethiopia”.

Subject: Submission of Revised manuscript (R1)

Dear esteemed editorial team (PLOS ONE Journal),

Many thanks for considering our manuscript for publication after revision. We are very much thankful to the critical review by the esteemed reviewers and academic editor. All the comments forwarded by the esteemed reviewers and academic editor are critically reviewed and improvement has been made in our revised manuscript. We provide a point-by-point discussion below and have updated the manuscript with the track changes. 

We would like also to thank the academic editor for considering our current situation (internet lockdown due to war) and extending the deadline for submitting our revised manuscript

I. Response to Academic editor

• Line 97 to 103, the subheading "Study area and population" was changed to “Study area description” (line 102 current version) and detailed description of the study area was given. The sampling techniques and reference population description were moved to the "Study design, Study Population and Sample size determination" subheading.

• Line 119 previous versions and line 141 current versions, representation for “Z” have been modified as per the editor comments.

• Description regarding the multivariate regression analysis technique was given focusing on model fitting, model building and quality parameters. The interpretation and presentation of odds ratio results is made according to standards and previous literature.

• The abbreviations used for zones were described in all tables and elsewhere

• Table on page 16 previous versions and page 18 on current version was edited as Table 4.

II. Response to Journal Requirements:

1. Please ensure that your manuscript meets PLOS ONE's style requirements, including those for file naming:

Response: The manuscript is prepared according to the ‘Manuscript Body Formatting Guidelines’ of PLOSE ONE

2. Survey or questionnaire related issue:

Response: Additional information on the questionnaire survey was included especially focusing on the pre-testing of the questionnaire. Moreover, the questionnaire used in this study is also attached as a supporting file (Line 167, current manuscript)

3. Please amend either the abstract on the online submission form or the abstract in the manuscript so that they are identical

Response: The abstract in the online submission form and manuscript are made identical

4. Please include a copy of Table 4 which you refer to in your text on page 12 and 13

Response: A copy of table 4 was included in the original submission; however, due to typing error it was labeled as Table 3. Now a correction is made accordingly.

5. Please include captions for your Supporting Information files at the end of your manuscript, and update any in-text citations to match accordingly

Response: Caption for supporting information files have been added at the end of the manuscript and in-text citations are also cross-checked.

6. Figure 1 copyright issue:

Response: The map of the study area presented in this manuscript is not copyrighted. We obtained shape files of ‘Ethio-region’, ‘Districts of Tigray region’, ‘digital elevation model (DEM)’ and ‘road network’ from the data center of the Institute of Climate and Society of Mekelle University and developed the study area map using the applications of ArcGIS. If required we have all the shape files and the ArcGIS file which we can submit for your reference.

III. Response to Reviewer #1:

• The language edition was made by a native English speaker (Dr. Matthew, T., associate Professor, College of Veterinary Medicine, Iowa State University), and he has acknowledged in the current version (444-445). 

• All the editorial and grammatical issues commented by the reviewers’ on line-by-line bases were addressed as follows:

Response to comment 1: Line 57 (previous version): the word ‘Taenia’ was misspelt and correction has been made according to the recommendation of the reviewer as ‘taeniasis/cysticercosis’ (Line 63 in the current version).

Response to comment 2: Line 108 (previous version) and line 123 (current version): The word “that” is replaced with “at” as per the suggestion.

Response to comment 3: Line 130 (previous version) and line 153 (current version): Table 1. Sample size for South-eastern zone is 40 (37 + 3) and 39 was written by typing error. When all the sample sizes are summed considering 40 for S/Eastern zone, it comes 421 which was our sample size. 

Response to comment 4: Line 301 (previous version) and line 362-63 (current version): FGD is written in full form.

Response to comment 5: Line 303 (previous version) and line 364 (current version): A full stop is added between Region and Anthrax. The sentence staring with "Anthrax" is also rephrased (line 364 –65).

Response to comment 6: Line 306 (previous version) and line 366-68 (current version): …………..public reluctance to vaccinate their animals. The vaccination for anthrax and other major diseases in Ethiopia are coordinated by the government and there is also a significant subsides for vaccines (e.g. the cost of Anthrax vaccine per animal is around 0.01 USD). The reluctance of farmers to vaccinate their animals’ is could be due to lack of awareness on the importance of vaccine.

Response to comment 7: Line 314 (previous version) and line 376 (current version): M. bovis is written in full and thank you for your critical observation

Response to comment 8: Line 329 (previous version) and line 391 (current version): the word “and” between Ethiopia and endemic is replaced with “as”.

Response to comment 9: Line 330 to 331 (previous version) and line 392–94 (current version): The sentence beginning “In the present study, brucellosis ………………in other studies [12, 21, 22] is rephrased to become understandable by the reader.

Response to comment 10: Line 335 to 338 (previous version) and line 398-403 (current version): This paragraph is entirely rephrased and improved. Sorry for the confusion. 

Response to comment 11: Line 345 (previous version) and line 412 (current version): …”fact that” is deleted from the sentence.

IV. Response Reviewer #2:

Response to comment 1: In addition to Table 2, barplots for differences in disease type by profession and experience of professionals are added as supporting files (S1 Fig and S2 Fig) and in-text description is also provided.

Response to comment 2: line 86 (previous version) and line 92 (current version): 'a' added before 'one health approach'

Response to comment 3: lines 108-109 (previous version) and line 123-25 (current version): use of capital letter for animals was changed into small letters

Biruk Mekonnen (corresponding author), on behalf of all co-authors

---

## [Decision Letter · Decision Letter 1]

7 Jun 2021

PONE-D-20-30909R1

Prioritization of Neglected  Tropical Zoonotic Diseases: A One Health Perspective from Tigray Region, Northern Ethiopia

PLOS ONE

Dear Dr. Mekonnen,

Thank you for submitting your manuscript to PLOS ONE. After careful consideration, we feel that it has merit but does not fully meet PLOS ONE’s publication criteria as it currently stands. Therefore, we invite you to submit a revised version of the manuscript that addresses the points raised during the review process.

Please attend to these minor reviews that have been recommended by one of the reviewers.

We look forward to receiving your revised manuscript.

Kind regards,

Martin Chtolongo Simuunza, PhD

Academic Editor

PLOS ONE

Journal Requirements:

Reviewers' comments:

Reviewer's Responses to Questions

**Comments to the Author**

1. If the authors have adequately addressed your comments raised in a previous round of review and you feel that this manuscript is now acceptable for publication, you may indicate that here to bypass the “Comments to the Author” section, enter your conflict of interest statement in the “Confidential to Editor” section, and submit your "Accept" recommendation.

Reviewer #1: All comments have been addressed

Reviewer #2: (No Response)

2. Is the manuscript technically sound, and do the data support the conclusions?

Reviewer #1: Yes

Reviewer #2: Yes

3. Has the statistical analysis been performed appropriately and rigorously? 

Reviewer #1: Yes

Reviewer #2: Yes

4. Have the authors made all data underlying the findings in their manuscript fully available?

Reviewer #1: Yes

Reviewer #2: Yes

5. Is the manuscript presented in an intelligible fashion and written in standard English?

Reviewer #1: Yes

Reviewer #2: Yes

6. Review Comments to the Author

Reviewer #1: There is need to thorough go through the document to clean some typing errors. e.g;

Some minor typographical comments in the manuscript:

Line 42: Delete “s” on challenges

Line 156: Delete “were” between “Who” and “already”

Line 279: Add “and” between production and productivity

Line 287: Delete the comma after “control”

Reviewer #2: Nearly all comments have been adequately addressed. Table 1 has a minor formatting error in title of third column ni=Ni*n/N) In re-reviewing the discussion, however, I think the authors should add a brief paragraph on study limitations -e.g. circumstances/available data that limited inferences drawn from the study.

7. PLOS authors have the option to publish the peer review history of their article (what does this mean?). If published, this will include your full peer review and any attached files.

Reviewer #1: No

Reviewer #2: No

---

## [Author Response · Author response to Decision Letter 1]

14 Jun 2021

Ref: PONE-D-20-30909R1

Title: Prioritization of Neglected Tropical Zoonotic Diseases: A One Health Perspective from Tigray Region, Northern Ethiopia

RE: Authors’ responses to reviewers/editors. 

Dears,

Thank you for the positive response to our manuscript. We are grateful for the insightful comments of both reviewers and editor. We have addressed the issues raised and proved point by point below and also updated the manuscript with track changes. 

1. Editor’s comment: Journal requirements on reference lists:

Answer: We have reviewed the reference list and found it complete and correct.

2. Reviewer #1 comments:

Comment 1: Line 42: Delete “s” on challenges

 Revised: Line 42: “… challenge by human health professionals”

Comment 2: Line 156: Delete “were” between “Who” and “already”

 Revised: Line 42: “… experts who already participated…”

Comment 3: line 279: Add “and” between production and productivity

 Revised: Line 279: “… low production, and productivity…”

Comment 4: line 287: Delete the comma after “control”

 Revised: Line 279: “… treatment, control and prevention programs…”

3. Reviewer #2 comments:

Comment 1: Table 1 has a minor formatting error in title of third column ni=Ni*n/N) 

 Revised: Line 133: it’s done. 

Comment 2: the authors should add a brief paragraph on study limitations -e.g. circumstances/available data that limited inferences drawn from the study.

Revised: Line 396-405: “This study presented some limitations: First, the absence of context specific to one health zoonotic disease prioritization tool could have influenced the prioritization of diseases. In addition, the lack of country-specific data for the majority of the zoonotic diseases has made it difficult to triangulate the results obtained in this study. Second, despite the fact those participants were from diversified disciplines, the number of animal health experts was fewer than human health experts, which may result in bias on professional opinions. In addition, the difference in the background of professionals involved in the study might have influenced the weighing and scoring results of zoonotic diseases. Finally, only 15 accessible districts which had a better road infrastructure and health facility were considered in this study which may not infer the non-accessible districts of the region.”

---

## [Editor Report · Decision Letter 2]

21 Jun 2021

Prioritization of Neglected  Tropical Zoonotic Diseases: A One Health Perspective from Tigray Region, Northern Ethiopia

PONE-D-20-30909R2

Dear Dr. Mekonnen,

We’re pleased to inform you that your manuscript has been judged scientifically suitable for publication and will be formally accepted for publication once it meets all outstanding technical requirements.

Kind regards,

Martin Chtolongo Simuunza, PhD

Academic Editor

PLOS ONE
---

## [Editor Report · Acceptance letter]

13 Jul 2021

PONE-D-20-30909R2 

Prioritization of Neglected Tropical Zoonotic Diseases: A One Health Perspective from Tigray Region, Northern Ethiopia 

Dear Dr. Mekonnen Wolde:

I'm pleased to inform you that your manuscript has been deemed suitable for publication in PLOS ONE. Congratulations! Your manuscript is now with our production department. 

Kind regards, 

on behalf of

Dr. Martin Chtolongo Simuunza 

Academic Editor

PLOS ONE